# Exploring the Use of Solid Biofertilisers to Mitigate the Effects of *Phytophthora* Oak Root Disease

Aida López-Sánchez [1,†], Miquel Capó [1,2,*,†], Jesús Rodríguez-Calcerrada [1], Marta Peláez [1], Alejandro Solla [3], Juan A. Martín [1,4] and Ramón Perea [1,4]

1   Departmento de Sistemas y Recursos Naturales, Universidad Politécnica de Madrid,
    Ciudad Universitaria s/n, 28040 Madrid, Spain
2   Departamento de Biología, Universitat de les Illes Balears. Cra. Valldemossa Km 7,5 s/n, 07122 Palma, Spain
3   Faculty of Foresty, Institute for Dehesa Research (INDEHESA), Universidad de Extremadura,
    Avenida Virgen del Puerto 2, 10600 Plasencia, Spain
4   Centro para la Conservación de la Biodiversidad y el Desarrollo Sostenible (CBDS),
    Universidad Politécnica de Madrid, C/José Antonio Novais 10, 28040 Madrid, Spain
*   Correspondence: miquelcaposervera@gmail.com; Tel.:+34-971-17-33-46
†   These authors contributed equally to this work.

**Abstract:** Oak forests are facing multiple threats due to global change, with the introduction and expansion of invasive pathogens as one of the most detrimental. Here, we evaluated the use of soil biological fertiliser Biohumin® to improve the response of *Quercus ilex* L. to the soil-borne pathogen *Phytophthora cinnamomi* Rands by using one-year-old seedlings fertilised at 0, 12.5, and 25% concentrations of Biohumin® (*v/v*). Our hypothesis was that plant vigour and response to the pathogen would improve with Biohumin®. The effects of soil infestation and fertilisation were tested by assessing plant survival, growth, and physiology. The soil infested with *P. cinnamomi* negatively affected all the studied traits. We observed that a moderate concentration of Biohumin® (12.5%) increased plant survival. However, a high concentration (25%) reduced the survival compared with the control, probably as a result of the stress caused by both biotic (infection) and abiotic (soil toxicity) factors. Biohumin® at the highest concentration reduced the plant height-to-stem diameter ratio (H/D) and negatively affected plant biomass and physiological activity. Combined biofertilisation and infection induced synergistic negative effects in the leaf water potential compared with infection and fertilisation applied alone. A higher concentration of Biohumin® may favour pathogens more than plants. Further studies should explore the causes of the negative effect of the high concentration of Biohumin® observed here and evaluate if lower concentrations may benefit plant survival and physiology against soil pathogens.

**Keywords:** anthropogenic disturbances; mineral nutrition; ecological restoration; Mediterranean forests; plant pathogens; plant physiology

## 1. Introduction

The increase in anthropogenic disturbances during the last century has generated new sources of stress in plants [1]. For instance, water deficit and temperature increase associated with human-induced climate change [2,3] have favoured the arrival and spread of emerging diseases and pests and have originated some abiotic–biotic interactions with synergistic negative effects on plants [4,5]. During the seedling stage, plants are highly vulnerable to abiotic and biotic stress, which limits seedling establishment and the probability to reach the subsequent life stages [6–8]. Consequently, in the short term, seedling recruitment might be highly compromised by diseases [9–11]. This situation calls for applied research to find solutions that reduce plant stress and recruitment failure.

The Mediterranean oak-dominated systems are facing unprecedented levels of biotic and abiotic stresses, such as drought, soil-borne pathogens, and insect outbreaks [12–14].

*Phytophthora* diseases are currently a major threat to many oak forests and are becoming an important threat to tree conservation [15–17]. The impact of *Phytophthora* species is enhanced by some factors such as habitat loss, drought, mismanagement, and extreme climate events [13,18–20]. In particular, the soil-borne root oomycete *Phytophthora cinnamomi* Rands is one of the most harmful and widespread pathogens worldwide [21–23]. Holm oak (*Quercus ilex* L.), cork oak (*Q. suber* L.), and chestnut (*Castanea* spp.) are particularly susceptible to *P. cinnamomi* in their seedling stage [24], with *Q. ilex* being the most susceptible oak species to *P. cinnamomi*. The root rot caused by *P. cinnamomi* is exacerbated by the high frequency and intensity of droughts and high temperatures in the Mediterranean area [25,26]. Waterlogging in combination with subsequent water deprivation is the worst scenario for *Q. ilex* if soils are infested with *P. cinnamomi* [26]. A higher frequency of extreme rain events that saturate the soil might be particularly beneficial for *P. cinnamomi*, potentially boosting its soil density beyond any possible defence response of the susceptible hosts [27]. However, an average drier climate might imply suboptimal conditions for *P. cinnamomi* infections, allowing for a slower advance of the disease in invaded areas [27]. There are no accurate data on the incidence or extent of holm oak decline in Spain, although, in 2010, official sources estimated the holm oak loss at ca. 8000 ha per year [28].

Soil chemical properties influence plant susceptibility to *Phytophthora* spp. infection [29,30], and the use of fertilisers has been proposed to enhance plant performance against pathogens [31]. Several studies have compared the benefits of organic vs. inorganic fertilisers on plant physiology and growth against *Phytophthora* species [32]. However, studies have mostly focused on crops [31,33], and less attention has been paid to species from natural ecosystems and foundation tree species such as oaks (*Quercus* spp.). In general, soil N fertilisation increases the photosynthetic machinery and plant vigour, which might have a positive effect on plant resistance to diseases, either by decreasing the infection by pathogens or by increasing plant tolerance and recovery potential after infection [34]. Previous research has explored the possibility of elevating the calcium content in soil to suppress the chlamydospore viability of pathogenic oomycetes [35]. The use of biological liquid substrates containing organic N, organic P, S, Ca, Mg, and trace elements (Fe, Mn, B, Zn) successfully reduced oak defoliation in the adult tree layer of a mixed oak forest stand [36]. In contrast, disease tolerance could be negatively affected if fertilisers induce a reduction in root-to-shoot ratio, which exacerbates the soil water uptake restrictions caused by root rot [37]. To reduce the impact of *Phytophthora* species in oak, it is important to search for solutions based on soil organic and/or inorganic fertilisers.

In this study, we aimed to assess the effect of the soil biological fertiliser Biohumin® (BIOHUMIN Deutschland GmbH, Berlin, Germany) on the vulnerability of holm oak seedlings to *P. cinnamomi*. This fertiliser was selected because (i) it contains fluvic humic acids and other trace elements that would improve soil water retention and cationic exchange, and (ii) its solid structure facilitates its application in the field. Specifically, we aimed to (i) elucidate how plant physiology and growth benefit from the application of the biological fertiliser and (ii) compare the impact of *P. cinnamomi* on fertilised and no-fertilised plants. Our hypotheses were that (i) plant vigour would increase after using Biohumin®, and (ii) fertilised individuals would be less susceptible to *P. cinnamomi* infection in terms of survival, water uptake, and plant vigour, in comparison to unfertilised plants.

## 2. Materials and Methods

### 2.1. Description of the Product

The product Biohumin® is an organic soil fertiliser that contains 0.5% organic N, 0.3% P, 0.5% S, 1% Ca, 0.2% Mg, 0.3% trace elements (Fe, Mn, B, Zn), 55% organic substance, 20% humic fulvic acids, and water (Biohumin® Deutschland GmbH, Berlin, Germany). Biohumin® improves the availability of trace elements in the soil (Fe, Mn, B, Zn). Biohumin® has been included in the FIBL (Research Institute of Organic Agriculture, Switzerland) list of biofertilisers as a product suited to soil biological structuring. This product was used as a model of solid mix (organic and inorganic) fertiliser during the experiment.

### 2.2. Seedling Origin and Cultivation

Seedlings from five adult holm oak trees from a forest infested by *P. cinnamomi* were used [38]. The forest is a 515 ha woodland dehesa located 445 m.a.s.l. in the Province of Huelva, SW Spain (37°52′ N, 6°14′ W), managed for swine production. Its climate is Mediterranean pluviseasonal oceanic, characterised by hot and long dry summers, from May to October, and strong interannual (452–894 mm/year) and intra-annual rainfall variability (5–128 mm/month). The selected mother trees had similar age, size, ecological conditions, and no external crown damage.

In November 2019, 100 sound acorns of similar size (ca. 8–10 g) were collected from each of the five selected holm oak trees. Acorns had no signs of insect damage (i.e., no oviposition perforations or larvae exit holes) and successfully passed a viability test using a flotation method [39]. The acorns were germinated inside a plastic bag for 60 days at 5 °C in the Faculty of Forestry (Universidad Politécnica de Madrid, Madrid, Spain). When the radicle emerged, at the end of January, we selected 360 acorns for plantation in 3 dm$^3$ pots filled with a sterilised substrate of the same weight (1.6 kg).

We prepared three substrates (*n* = 360 seedlings; 120 for each soil treatment): (1) 25% silica sand, 75% peat NOVARBO C1LE 70/30D, NPK 12-14-24, Novarbo®, and 0% Biohumin® (hereafter BIO 0% (*v/v*) plants); (2) 25% sand, 50% peat, and 25% Biohumin® (BIO 25% (*v/v*) plants); and (3) 25% sand, 62.5% peat, and 12.5% Biohumin® (BIO 12.5% (*v/v*) plants). Percentages refer to the volumes used for each compound of the substrate and were chosen to compare our results with those reported in the previous literature that used a concentration of 18% [40]. For each soil treatment (*n* = 120 seedlings), half of the plants were assigned to the *P. cinnamomi* infection treatment group and the other half to a control group (Figure 1). On each pot, we introduced two 0.06 dm$^3$ cylindrical tubes to inoculate the plant with *P. cinnamomi* without damaging the root system [41]. Five months after seedling emergence, inoculation with *P. cinnamomi* was performed (see below).

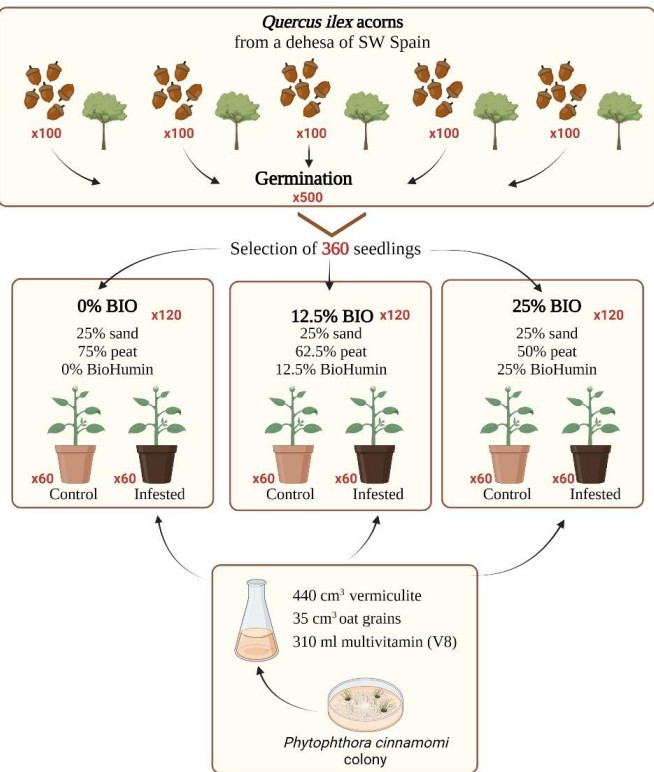

**Figure 1.** Experimental design including details of acorn sampling, soil fertilisation, acorn germination, plant growth, and *P. cinnamomi* soil infestation. Red numbers refer to the sample size at each step.

### 2.3. Phytophthora cinnamomi Cultivation and Inoculation

The *P. cinnamomi* A2 strain used in the experiment (code UEx1) was isolated from the rhizosphere of an infected *Q. ilex* tree located in Valverde de Mérida (Badajoz, Spain) [9]. The identity of the pathogen was checked by comparing its morphological features with those typical for *P. cinnamomi*. The inoculum was prepared following the methodology of Jung, Blaschke, and Neumann (1996), with some modifications [42]. We mixed 440 cm$^3$ of fine vermiculite, 35 cm$^3$ of whole oat grains, and 310 mL of multivitamin (V8) juice broth (containing 200 mL L$^{-1}$ of V8 juice and 800 mL L$^{-1}$ of demineralised water amended with 3 g L$^{-1}$ CaCO$_3$) in 1-l Erlenmeyer flasks that were autoclaved twice. Four 5 mm × 5 mm plugs from the border of a *P. cinnamomi* colony growing in Petri dishes with potato, dextrose, and agar were added to the flasks containing the medium and kept in the dark at 20 °C for five weeks. Additional flasks containing the same medium but without *P. cinnamomi* plugs were incubated as controls for application to BIO 0% plants.

Soil infestation was conducted on 24 June 2019. The inoculum was rinsed with demineralised water to remove excess nutrients. Then, the two cylinders per pot were removed to deliver the inoculum in the empty volume. Each plant received 120 mL of inoculum, while the control plants received the same quantity of culture medium without *P. cinnamomi*. To stimulate *P. cinnamomi* sporulation and zoospore release, the infested and control pots were waterlogged for 2 days by using two different pools (3300 dm$^3$). All the plants were under the same environmental conditions, in a greenhouse with automatic irrigation and cooling.

At the end of the experiment, *P. cinnamomi* was reisolated from the root samples collected from the artificially infested soil. Rootlets from six randomly chosen plants from each treatment were cut into 1 cm segments, surface-sterilised (1 min in 1% sodium hypochlorite), rinsed with sterile water, dried in a laminar flood chamber, and finally plated onto NARPH agar selective medium. The colonies of *P. cinnamomi* were identified under a microscope through their distinctive morphological structures such as clustered hyphal swellings, chlamydospores, and sporangia.

### 2.4. Response Variables

Oak survival was recorded every 10 days from 24 June to 3 September. The morphological and physiological parameters were assessed in September, at the end of the experiment, by using a subsample of 36 plants for biomass (6 plants × 3 soil treatments (ST) × 2 *Phytophthora* infection treatments (PI)) and 60 plants for size and physiology (10 plants × 3 soil treatments × 2 *Phytophthora* infection treatments (PI)).

As morphological variables, we measured the plant height (H), the basal diameter (D), and the H/D ratio. Thereafter, we harvested the plants and separated the organs for oven-drying (3–4 days at 72 °C). The dry biomass of fine roots (i.e., roots < 2 mm diameter), coarse roots, stems, and leaves were weighed to calculate the leaf mass fraction (LMF, g leaves g$^{-1}$ plant), the stem mass fraction (SMF, g stems g$^{-1}$ plant), the root mass fraction (RMF, g roots g$^{-1}$ plant), the fine-root mass fraction (FRMF, g fine roots g$^{-1}$ plant), the fine-root-to-leaf biomass fraction (FR/LB, g fine root g$^{-1}$ leaves), and the root-to-shoot ratio (R/S ratio, g roots g$^{-1}$ stems plus leaves).

As physiological variables, we measured the leaf water potential (Ψ), the leaf gas exchange, and chlorophyll *a* fluorescence. The water potential of one leaf per plant was measured at midday with a pressure chamber (PMS Instrument Company, Albany, OR, USA). The leaf gas exchange and chlorophyll *a* fluorescence were measured with a Li-6400 portable photosynthesis system (Li-Cor Inc., NE, USA). We used a 6400-40 leaf chamber fluorometer, which includes a 2 cm$^2$ leaf chamber equipped with low emission diodes providing both actinic and far-red light. The measurements were made in two days, approximately from 11:00 h to 14:00 h, in the leaves already acclimated to full sunlight in the greenhouse. We set the airflow at 300 μmol s$^{-1}$, the temperature at 25 °C, the light at 1200 μmol m$^{-2}$ s$^{-1}$, and the air CO$_2$ concentration at 400 ppm. The air's relative humidity ranged between 45% and 65% across the measurements. Under these conditions,

we measured net $CO_2$ assimilation ($P_n$), stomatal conductance to water vapor ($g_s$), transpiration I, and intercellular $CO_2$ concentration ($C_i$). The $P_n/g_s$ ratio was used as a proxy of the instantaneous water use efficiency. After recording the gas exchange variables, fluorescence was measured under "steady-state" light conditions (1200 µmol m$^{-2}$ s$^{-1}$; Fs) and after a saturating light pulse (>7000 µmol m$^{-2}$ s$^{-1}$; Fm') to calculate the effective quantum yield of photosystem II ($\Phi_{PSII}$: F–' − Fs)/Fm') and the electron transport rate (ETR: $\Phi_{PSII} \times 0.84 \times 0.5 \times 1200$ µmol m$^{-2}$ s$^{-1}$ [43].

### 2.5. Data Analysis

The maximal models (containing all the predictors) used for analyses are summarised in Table 1. For oak seedling survival, we used Kaplan–Meier non-parametric models (Table 1). In the maximal model, we clustered all the oak seedlings that originated from the same mother tree to indicate the correlated groups of observations within the survival analysis (Table 1).

**Table 1.** Summary of maximal models performed for data analysis in this study.

| Model Type | Model | Response Variable (Group) [1] | Fixed Effect [2] | Random Effect | Sample Size (n) |
|---|---|---|---|---|---|
| Kaplan–Meier | I | Oak survival | ST × PI | 1 \| Mother Tree | 360 |
| GLMMs | II | Morphology (biomass) | ST × PI | 1 \| Mother Tree | 36 |
| GLMMs | III | Morphology (size) | ST × PI | 1 \| Mother Tree | 60 |
| GLMMs | IV | Physiology | ST × PI | 1 \| Mother Tree | 60 |

[1] Morphology and physiology includes several response variables. [2] Fixed factors = ST: Biohumin® soil treatment (0% vs. 12.5% vs. 25% Biohumin® fraction); PI: *Phytophthora* soil infestation treatment.

For oak morphological and physiological variables, we developed generalised linear mixed models (GLMMs) [44]. Box–Cox transformations [45] were applied when needed to calculate the transformation lambda that maximises the likelihood function. Thus, some of the response variables were fitted to a gamma error distribution with their corresponding power lambda link function (Table S1). When monotonic transformations were not necessary, the response variables were fitted to a Gaussian error distribution with the identity function. For all the models, the analyses included the mother tree to account for the random effect structure (Table 1).

We used the model averaging approach to select the final models [46]. We first fitted the maximal model containing all the predictors. Then, we ranked all the possible models derived from the maximal model by their AIC weights using the "dredge" function within the "MuMIn" package of R and selected those with the best AIC weight (hereafter top models), i.e., which had ΔAIC < 2 (Table S2) [46]. Finally, we obtained the model-averaged coefficients of the top models as well as the relative importance of each predictor (from 0 to 1) by using the "model.avg" function of "MuMIn" (Table S3) [46].

Data processing and statistics were performed using R 3.6.0 [47] with the modules "lme4" [48], "car" [49], and "MuMIn" [50].

## 3. Results

### 3.1. Survival of Oak Seedlings

The infection of plants with *P. cinnamomi* affected seedling survival ($x^2$ = 13.700, $p < 0.001$; Figure 2A), which was higher in the control plants (80.5%) than in the infected ones (55.1%) at day 161 post-infestation. In contrast, similar values of seedling survival were found between soil treatments ($x^2$ = 0.600, $p = 0.800$; Figure 2B). The interaction between soil treatment and infection by *P. cinnamomi* was significant ($x^2$ = 19.500, $p = 0.002$), because seedling survival was similar in the control and infected plants using the 25% BIO treatment but different in 0 and 12.5% BIO treatments, with the control plants always surviving more than the infected ones.

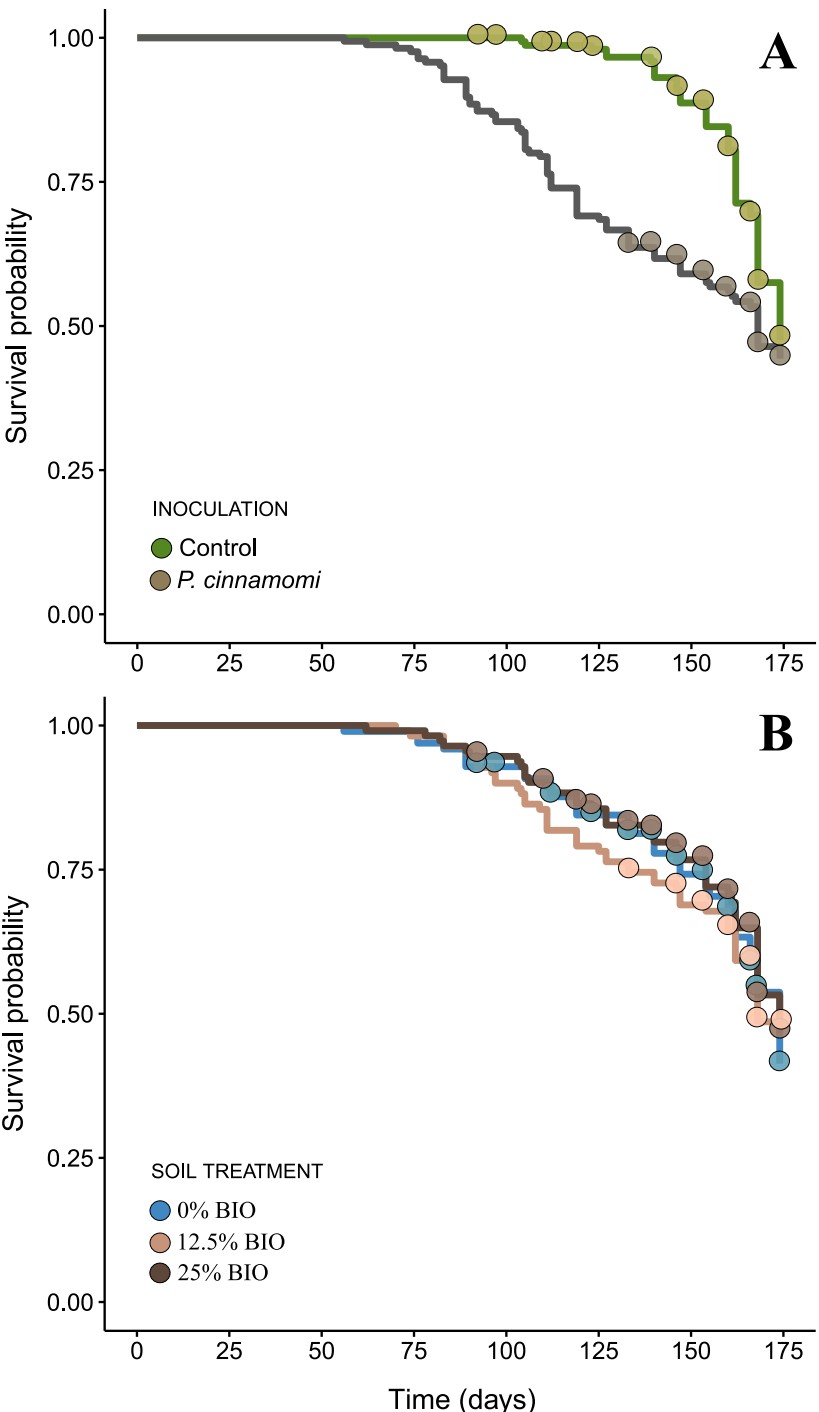

**Figure 2.** Survival probability of *Q. ilex* plants since plant emergence, depending on the infection by *P. cinnamomi* (**A**) using mock-inoculated plants vs. *P. cinnamomi*-inoculated plants, and the Biohumin® concentration treatments (**B**) with 0, 12.5, and 25% BIO proportion in substrate. Dots indicate measures taken during experiment, and lines are the predicted values by the model.

### 3.2. Effects on Plant Growth

Treatments with 12.5% BIO and 25% BIO showed lower values of LMF (≈0.84-fold difference) than 0% BIO (Table 2, Figure 3). In addition, the plants infected with *P. cinnamomi* had higher LMF than the controls when using 12.5% BIO and 25% BIO soil treatments (1.26- and 1.14-fold difference, respectively), while the opposite was observed in 0% BIO (Figure 3).

**Table 2.** Summary of the top linear mixed models ($\Delta$AIC < 2) fitted to analyse main and interaction effects of factors soil treatment (ST) and *P. cinnamomi* infection (PI) on morphological variables (II and III models in Table 1). Asterisks indicate statistical significance ($p < 0.05$ *, $p < 0.01$ **, $p < 0.001$ ***).

| Response Variable | Fixed Effects | Importance [2] | Levels | Coeff. | SE | z-Value | p |
|---|---|---|---|---|---|---|---|
| Root biomass [1] | Intercept | | | 0.576 | 0.050 | 11.440 | 0.001 *** |
| LMF | Intercept | | | 26.743 | 2.056 | 12.633 | <0.001 *** |
| | Soil Treatment | 0.85 | Bio 12.5% | −7.089 | 3.235 | 2.142 | 0.032 * |
| | | | Bio 25% | −4.570 | 2.942 | 1.511 | 0.031 * |
| | PI Infection | 0.60 | *P. cinnamomi* | −2.277 | 3.421 | 0.651 | 0.515 |
| | ST × PI | 0.39 | B 12.5% × PI | 8.600 | 3.945 | 2.092 | 0.036 * |
| | | | B 25% × PI | 6.700 | 3.945 | 1.630 | 0.003 ** |
| FRMF | Intercept | | | 9.496 | 0.862 | 10.674 | <0.001 *** |
| | PI Infection | 0.40 | *P. cinnamomi* | −1.351 | 1.337 | 0.974 | 0.330 |
| R/S ratio | Intercept | | | 1.148 | 0.035 | 31.339 | <0.001 *** |
| | Soil Treatment | 0.72 | Bio 12.5% | 0.060 | 0.029 | 1.969 | 0.050 |
| | | | Bio 25% | 0.031 | 0.036 | 0.833 | 0.405 |
| | PI Infection | 0.94 | *P. cinnamomi* | −0.049 | 0.028 | 1.717 | 0.086 |
| | ST × PI | 0.23 | B 12.5% × PI | −0.011 | 0.051 | 0.211 | 0.833 |
| | | | B 25% × PI | −0.079 | 0.051 | 1.473 | 0.141 |
| FR/LB | Intercept | | | 9.229 | 0.669 | 13.800 | <0.001 *** |
| Height (H) | Intercept | | | 20.209 | 1.196 | 16.539 | <0.001 *** |
| | Soil Treatment | 0.84 | Bio 12.5% | −2.205 | 1.595 | 1.353 | 0.176 |
| | | | Bio 25% | −4.229 | 1.577 | 2.625 | 0.009 ** |
| | PI Infection | 0.33 | *P. cinnamomi* | −0.026 | 1.295 | 0.020 | 0.984 |
| Basal diameter (D) | Intercept | | | 4.510 | 0.161 | 27.413 | <0.001 *** |
| | PI Infection | 0.29 | *P. cinnamomi* | 0.251 | 0.265 | 0.926 | 0.354 |
| H/D ratio | Intercept | | | 4.907 | 0.404 | 11.947 | <0.001 *** |
| | Soil Treatment | 0.88 | Bio 12.5% | −0.817 | 0.454 | 1.762 | 0.078 |
| | | | Bio 25% | −1.340 | 0.569 | 2.325 | 0.020 * |
| | PI Infection | 0.52 | *P. cinnamomi* | −1.077 | 0.497 | 2.116 | 0.034 * |
| | ST × PI | 0.36 | B 12.5% × PI | 0.738 | 0.693 | 1.039 | 0.059 |
| | | | B 25% × PI | 1.524 | 0.685 | 2.170 | 0.160 |

[1] Biomass of roots less than 2 mm in diameter. LMF: leaf mass fraction; FRMF: fine-root mass fraction; R/S ratio: root-to-shoot ratio; FR/LB: fine-root-to-leaf biomass fraction.. Results for soil treatment refer to Biohumin® 12.5% and 25% against control plants (i.e., Biohumin® 0% is included in intercept), and results for biotic stress refer to *P. cinnamomi*-infected plants against control plants (i.e., uninfected plants are included in intercept). [2] *Importance*: Importance of predictor variable in the model averaging approach.

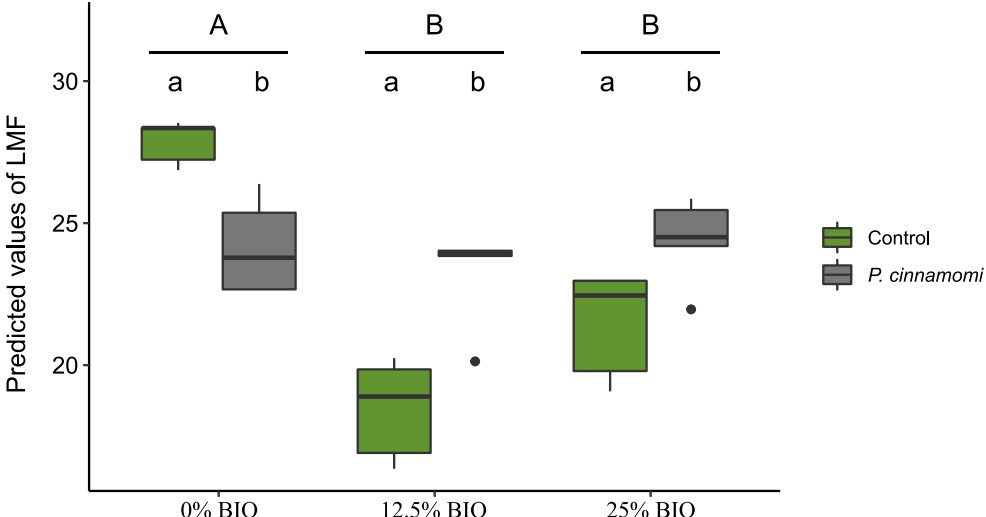

**Figure 3.** Boxplot of predicted values of leaf mass fraction (leaf dry mass/total plant dry mass; LMF) separated by soil treatments (*x*-axis) and *P. cinnamomi* infection (control plants shown in green colour; infected plants in grey). Lowercase letters indicate differences between *P. cinnamomi* infection treatments, and capital letters indicate differences among soil treatments.

No differences were observed between soil treatments and *P. cinnamomi* treatments for RMF or R/S. However, the plants treated with 25% BIO had lower heights and H/D ratios than those treated with 0% BIO (0.79- and 0.80-fold difference, respectively; Table 2). There were no significant differences in the height ($Z = 1.273$, $p = 0.203$) and the H/D ratio ($Z = 1.091$, $p = 0.275$) between the 12.5% BIO- and 25% BIO-treated plants. In addition, the plants infected with *P. cinnamomi* had lower H/D ratios than the mock-inoculated plants in 0% BIO and 12.5% BIO but not in 25% BIO (0.97-fold difference for 0% BIO and 0.92-fold difference for 12.5% BIO, respectively; Figure 4).

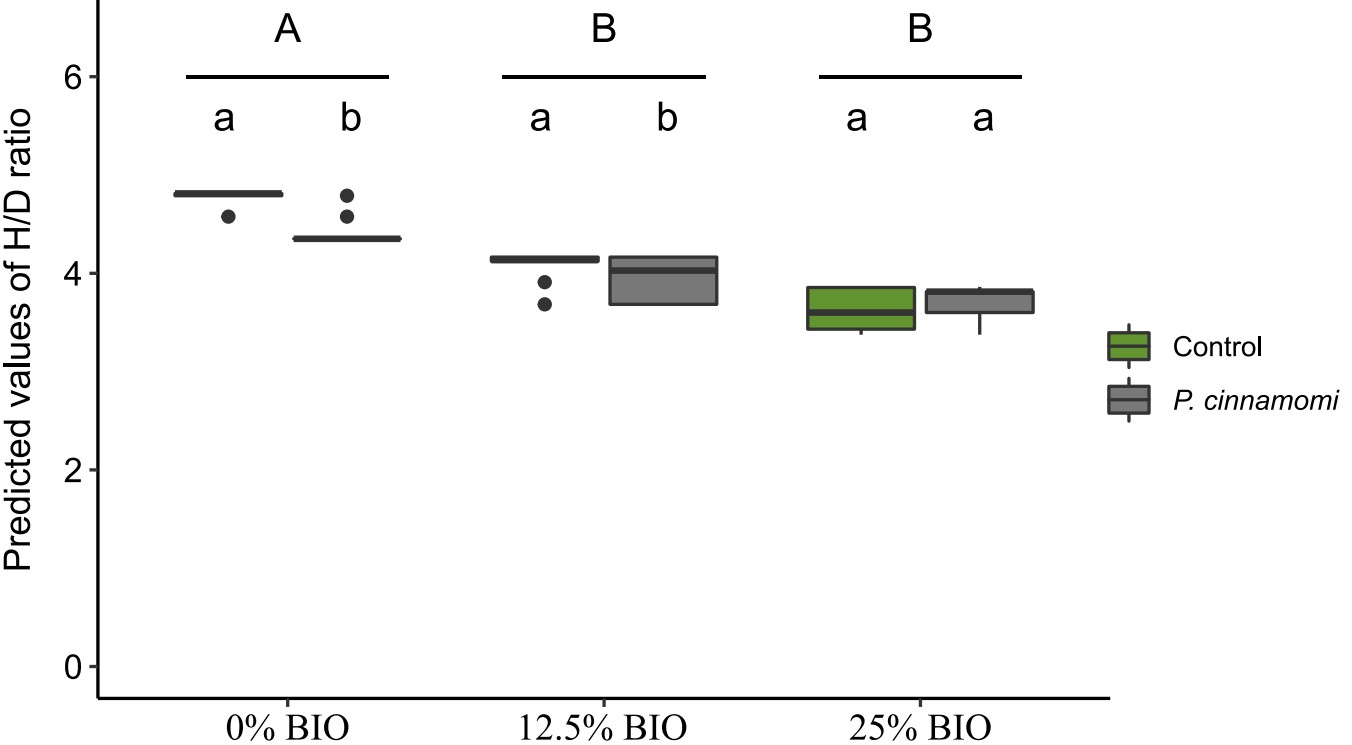

**Figure 4.** Boxplot of predicted values of height:diameter ratio (H/D) separated by soil treatments (*x*-axis) and *P. cinnamomi* infection (control plants, green; infected plants, grey). Lowercase letters indicate differences between *P. cinnamomi* infection treatments, and capital letters indicate differences among soil treatments.

### 3.3. Effects on Plant Physiology

We did not find any significant differences in $P_n$, $g_s$, $C_i$, $E$, $P_n/g_s$, or $P_n/E$ concentrations between soil treatments and *P. cinnamomi* treatments (Table 3). However, the *P. cinnamomi*-infected plants showed marginally higher $C_i$ than the control plants (1.05-fold difference; $p = 0.064$; Table 3), and marginally lower $P_n/g_s$ (0.87-fold difference; $p = 0.089$; Table 3). We found significant differences in the ETR and $\Psi$ depending on soil treatments (Table 3). The fertilised plants (12.5% BIO and 25% BIO treatments) showed lower values of the ETR and $\Psi$ ($\approx 0.76$ and $\approx 0.79$-fold difference, respectively) than the plants not fertilised (0% BIO; Table 3). In addition, *P. cinnamomi*-infected plants showed more negative $\Psi$ (1.27-fold difference) than the control plants (Table 3).

We found no significant interaction between soil treatments and *P. cinnamomi* treatments for any of the physiological variables. Interactions did not appear in the top models during model selection due to their low relative importance in modelling (Table S4).

**Table 3.** Summary of the top linear mixed models (ΔAIC < 2) fitted to analyse main factors—soil treatment and *P.c.* infection—affecting physiological variables (IV models of Table 1). Asterisks indicate statistical significance ($p < 0.05$ *, $p < 0.01$ **, $p < 0.001$ ***).

| Variable | Fixed Effects | *Importance* [2] | Levels | *Coeff.* | *SE* | *z-Value* | *p* |
|---|---|---|---|---|---|---|---|
| $P_n$ | Intercept | | | 8.476 | 0.615 | 13.543 | <0.001 *** |
| | *P.c.* Infection | 0.44 | *P. cinnamomi* | −1.045 | 0.941 | 1.087 | 0.277 |
| $g_s$ | Intercept | | | −1.834 | 0.109 | 16.521 | <0.001 *** |
| | Soil Treatment | 0.31 | Bio 12.5% | −0.263 | 0.18081 | 1.424 | 0.154 |
| | | | Bio 25% | −0.180 | 0.18131 | 0.969 | 0.332 |
| | *P.c.* Infection | 0.32 | *P. cinnamomi* | −0.089 | 0.147 | 0.591 | 0.554 |
| $C_i$ | Intercept | | | 284.927 | 5.609 | 50.063 | <0.001 *** |
| | *P.c.* Infection | 0.69 | *P. cinnamomi* | 13.514 | 7.137 | 1.854 | 0.064 |
| ETR | Intercept | | | 1.085 | 0.002 | 645.189 | <0.001 *** |
| | Soil Treatment | 0.96 | Bio 12.5% | −0.005 | 0.002 | 2.613 | 0.009 ** |
| | | | Bio 25% | −0.006 | 0.002 | 2.878 | 0.004 ** |
| | *P.c.* Infection | 0.33 | *P. cinnamomi* | 0.001 | 0.002 | 0.305 | 0.761 |
| E | Intercept | | | 1.941 | 0.150 | 12.973 | 0.001 ** |
| $P_n/g_s$ ratio | Intercept | | | 61.364 | 3.384 | 17.866 | <0.001 *** |
| | *P.c.* Infection | 0.63 | *P. cinnamomi* | −7.667 | 4.423 | 1.697 | 0.089 |
| $P_n/E$ ratio | Intercept | | | 4.354 | 0.202 | 21.593 | 0.001 *** |
| Ψ | Intercept | | | 3.004 | 0.095 | 31.700 | <0.001 *** |
| | Soil Treatment | 0.85 | Bio 12.5% | −0.216 | 0.095 | −2.270 | 0.023 * |
| | | | Bio 25% | −0.232 | 0.096 | −2.413 | 0.016 * |
| | *P.c.* Infection | 0.97 | *P. cinnamomi* | 0.236 | 0.076 | 3.108 | 0.002 ** |

$P_n$: Net $CO_2$ assimilation; $g_s$: stomatal conductance to water vapor; $C_i$: intercellular $CO_2$ concentration; ETR: electron transport rate; E: transpiration; $P_n/g_s$ ratio: proxy of instantaneous water use efficiency; Ψ: water potential. *Importance*: Importance of predictor variable in the model averaging approach. Results for soil treatment refer to Biohumin® 12.5% and 25% against control plants (Biohumin® 0%), and results for biotic stress refer to *P. cinnamomi*-infected plants against control (uninfected) plants. [2] *Importance*: Importance of predictor variable in the model averaging approach.

## 4. Discussion

Several studies reported how biological fertilisers favour plant growth and physiology [51–53]. However, the role of biological fertilisers in enhancing plant resistance against soil-borne pathogens is not clear and is still under study. Assessing the combination of biological fertilisers and *P. cinnamomi*-soil infection is crucial to disentangle whether biological fertilisers are positive for plants, or if they induce plant stress. Here, we observed that the Biohumin® fertiliser affected plant susceptibility to *P. cinnamomi*, with high concentrations causing additional stress to soil infestation.

### 4.1. Survival of Oak Seedlings

Soil infestation by *P. cinnamomi* compromises the proper development of seedlings and limits the recruitment of tree populations [22,23], including *Quercus* species [6,38,54]. Our results showed that the survival of the *Q. ilex* seedlings infested by *P. cinnamomi* was lower than that of the control plants, in agreement with previous studies [9,55]. This negative effect has been observed in *Q. suber* forests [56], and it is considered an important threat to oak conservation in Mediterranean systems [56–58].

It was expected that the plant survival after *P. cinnamomi* inoculation would increase with Biohumin® treatment, as reported for other inorganic and organic fertilisers [59,60]. In this study, it was observed that the plants fertilised with 12.5% Biohumin® or not fertilised had higher survival rates than the plants fertilised with 25% Biohumin®, across all the non-infested treatments. However, the highest concentration of Biohumin® (25% Bio) did not show a positive effect on the survival of the infected plants, compared with that of the uninfected plants. It appears that a combined biotic (infection) and abiotic (soil toxicity)

stress occurred, as reported in other studies [7,61,62]. By exploring the plant growth and leaf physiological parameters in this study, we sought to shed light on the effect of the interaction between fertilisation application and *P. cinnamomi* infection on plant survival.

### 4.2. Plant Growth

Plant growth was reduced after the inoculation of *P. cinnamomi* depending on the dosage of Biohumin®. Previous studies have reported that *P. cinnamomi* negatively affects plant growth and development [21,23]. Our results showed that the inoculation of *P. cinnamomi* reduced the fraction of the biomass invested in leaves (LMF), the plant height (H), and the ratio of plant height-to-stem diameter (H/D). This effect is not rare, as many pathogens from the genus *Phytophthora* spp. reduced the aerial biomass (leaves and stems) or even belowground parts [41,63].

On the other hand, the use of a high concentration of Biohumin® (25% Bio) influenced plant growth; it reduced the plant height (but not the stem diameter) and H/D. This result reinforces the idea that 25% of Biohumin® was excessive and negative for plant development and, therefore, did not protect the plants against *P. cinnamomi*. In fact, the plants treated with 25% of Biohumin® showed a similar plant size irrespective of *P. cinnamomi* infection. It is not uncommon that high concentrations of fertilisers interfere with the correct development of plants, as reported for inorganic [64] and organic fertilisers [65].

### 4.3. Plant Physiology

The impact of *P. cinnamomi* infection was also noticeable on the physiology of plants. The variables ETR and Ψ were lower after the inoculation of *P. cinnamomi*. After root infection, *P. cinnamomi* grows inter- and intracellularly in the host tissue, causing severe structural changes in *Q. ilex* [66,67]. A common consequence of root destruction by *Phytophthora* spp. is the decreased water absorption capacity of the infected plants. For example, root water transport failure was observed on susceptible *Eucalyptus sieberi* when infected with *P. cinnamomi* [68]. A major reduction in hydraulic conductivity was observed within the first 2 weeks after infection, although the pathogen had colonised only 8%–12% of the total root system, indicating that the infection by *P. cinnamomi* of a susceptible host could trigger a generalised dysfunction in plant–water relations, which could be mediated by hormonal changes and the toxins released by the pathogen [69]. In this study, the ETR and Ψ did not decrease to the levels jeopardising survival, which probably reinforces the assumption that other factors are involved during infection. The results presented here revealed evidence indicating that the pathogen negatively affected the plant function, reducing the plants' biomass increment and reallocation of resources, as observed in other studies [41].

Biofertilisation also reduced the ETR and Ψ, independently of the concentration used in the substrate. Although the effects were of low magnitude, they indicated that the use of Biohumin® at high concentrations did not help improve the physiology of the plants; rather, it may have caused toxicity [65]. Furthermore, the combined fertilisation and *P. cinnamomi* soil infestation induced synergistic effects in Ψ, causing more stress for plants than when using each factor alone. Further studies must evaluate if concentrations lower than those applied in this study may benefit plant physiology.

### 4.4. Using Biological Fertilisers to Enhance Protection against P. cinnamomi

Overall, the use of Biohumin® did not seem to help the plants to cope with soil infestation, and some variables indicated low plant vigour when *P. cinnamomi* inoculation and fertilisation were combined. Contrary to our expectations, the 25% concentration of Biohumin® negatively affected the *Q. ilex* individuals, probably due to plant toxicity, as discussed above. The use of fertilisers to reduce plant susceptibility to disease is under debate, but many positive effects have been reported, especially on crops [70,71]. Hence, the use of a particular fertiliser must be experimentally supported in each circumstance. According to our experiment, in terms of many of the study variables, the use of Biohumin®

was not beneficial to promote plant resistance against *P. cinnamomi*, especially when applied at high concentrations (i.e., 25% BIO).

## 5. Conclusions

In this study, the biological fertiliser Biohumin® at 12.5% (*v*/*v*) increased the survival of one-year-old *Q. ilex* seedlings after inoculation with *P. cinnamomi*. This advantage was not observed when using the fertiliser at 25% (*v*/*v*). At this higher dose, the combined effect of *P. cinnamomi* infection and fertilisation had negative, synergistic effects on plant survival and vigour. Plants showed low physiological activity after infection by *P. cinnamomi* and the application of fertiliser at 25% (*v*/*v*). Revealing the proper dose of biological fertilisers is relevant to improve plant resistance to *P. cinnamomi* and favour tree regeneration and sustainability in oak forests.

**Supplementary Materials:** The following supporting information can be downloaded at: https://www.mdpi.com/article/10.3390/f13101558/s1, Table S1: Summary of evaluated morphological and physiological response variables for data analysis; Table S2: Summary of Kaplan–Meier non-parametric models fitted to analyse the main factors—soil treatment and *P.c* Infection—affecting the survival of oak seedlings; Table S3. Results of model selection of the generalised linear mixed models fitted to analyse the main factors—soil treatment (S), biotic stress (B)—affecting different morphological response variables; Table S4. Results of model selection of the generalised linear mixed models fitted to analyse the main factors—soil treatment (S), biotic stress (B)—affecting different physiological response variables.

**Author Contributions:** Conceptualisation, R.P., J.A.M. and A.L.-S.; methodology, A.L.-S., J.R.-C., J.A.M., M.P. and A.S.; formal analysis, M.C. and A.L.-S.; resources, R.P. and A.L.-S.; writing—original draft preparation, M.C., A.L.-S, R.P. and J.R.-C.; writing—review and editing, all authors; supervision, J.A.M., A.S., R.P. and A.L.-S.; funding acquisition, R.P. and A.L.-S. All authors have read and agreed to the published version of the manuscript.

**Funding:** This research was funded by Universidad Politécnica de Madrid and Comunidad de Madrid, through project GLOBAFLOR Ref. APOYO-JOVENES-9OHUU0-10-3L226X (pluriannual agreement for young researchers).

**Data Availability Statement:** Not applicable.

**Acknowledgments:** We thank Fundación Monte Mediterráneo, and especially Ernestine Lüdeke and Hans-Gerd Neglein, for facilitating the acorn collection. We also thank David Medel, Jorge Pallarés, Eva Miranda, and Luis Adrián Lara for their help in lab work. M.C. and M.P. were supported by Margarita Salas postdoctoral fellowship (Ministerio de Universidades, Gobierno de España) through the Recovery, Transformation, and Resilience Plan funded by the European Union (NextGenerationEU).

**Conflicts of Interest:** The authors declare no conflict of interest.

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
