# Peer review of "Exploring the Use of Solid Biofertilisers to Mitigate the Effects of Phytophthora Oak Root Disease"

_forests, doi:10.3390/f13101558_

Round 1

Reviewer 1 Report

General Comments

The research paper investigates the use of solid biofertilizers to mitigate the effects 2 of Phytophthora oak root disease under greenhouse conditions. The paper is very well written, the theme is of high interest, and authors provided interesting data. Although, before being considered for publication some clarifications and additional data should be provided. There is several types of biofertilizers some of which are more adequate to alleviate biotic and abiotic stress. The specific choice of humic sbstances rich product (including dose of application) should be justified (specifically for a forest ecosystem). The discussion section should be improved and should state elaborate research hypothesis and not be limited to example of previous results.      

Specific comments:

In the abstract and introduction section, a relevant research hypothesis is lacking. There is several biofertilizers in the market with various compositions. The choice of Biohumin (rich in humic substances) is motivated by what? It is not clear why the author studied the effect of this specific biofertilizer, knowing that there are several products that might better target Phytophthora infection

In the introduction section example of real impact of pathogens on oak should be provided. What is the eco-environmental impact and how this justifies the study.  

Line 75-77: the author hypothesizes that the infection should be alleviated through having better plant vigor and growth. This is expected to be a result of the application of biofertilizers. How is this different from the application of chemical fertilizers? What is the added value of biofertilizers (which is expected to provide only nutrient In this specific case) in a forest ecosystem where there is enough organic matter and humic substances?  

Line 106-107. Why specifically 12.5% and 25% of “Biohumin”? Is it based on manufacturer recommendations or plant nutrient requirement? Dose of application of the organic fertilizer should be justified  

Line 115: Provide the accession number (was it identified?) of P. cinnamomic

Line 143-179 (response variable): Only basic physiological traits were measured (photosynthetic parameter and stomatal conductance…). Plants were subjected to biotic stress. The study will highly benefit from studying changes in enzyme related stress (SOD, CAT, APX, GPX…) in stressed and non-stressed plants.

Line 273: what do you mean by controversial? Provide references and further develop the idea

Line 291: you are attributing lower plant growth to infection but also toxicity. This is not clear and requires further investigation. The toxicity is due to what (composition wise)? Chemical analysis of your substrate before and after harvesting may provide clearer indications. Moreover, it was evident that by increasing the organic fertilizer concentration plant growth was negatively affected. You state that this was expected. The question here is why did you not assay lower concentrations (between 0 and 10% for example)?

Reviewer 2 Report

The manuscript titleg “Exploring the use of solid biofertilizers to mitigate the effects of Phytophthora oak root disease” investigating impact of the soil-borne pathogen Phytophthora cinnamomi on the Mediterranean oak tree. This is a very meaningful work, and the research ideas are very clear. The methods used are appropriate and advanced. However, it is not perfect and needs minor revision.

1.       Why the authors choose the specific product “fertilizer Biohumin” (BIOHUMIN Deutschland GmbH, Berlin, Germany), do you have any reason?

2.       Figure 3 and 4à how did you compare the means? What method did you use? Explain it.

3.       The authors use v/v in conclusion, you must give explicit to it in introduction or use the complete form in conclusion.

4.       What is the implication of your study? It is obvious that you cannot use fertilizer in large scales like forest.
